# Nanocrystalline Cubic Phase Scandium-Stabilized Zirconia Thin Films

**DOI:** 10.3390/nano14080708

**Published:** 2024-04-18

**Authors:** Victor Danchuk, Mykola Shatalov, Michael Zinigrad, Alexey Kossenko, Tamara Brider, Luc Le, Dustin Johnson, Yuri M. Strzhemechny, Albina Musin

**Affiliations:** 1Department of Chemical Engineering, Biotechnology and Materials, Faculty of Engineering, Ariel University, Ariel 40700, Israel; danchuk@vut.cz (V.D.); nikopolshade@gmail.com (M.S.); zinigradm@ariel.ac.il (M.Z.); kossenkoa@ariel.ac.il (A.K.); 2CEITEC BUT, Brno University of Technology, 61200 Brno, Czech Republic; 3Raicol Crystals Ltd., Rosh Ha’Ayin 4809162, Israel; 4Technology and Engineering Unit, Electronic Microscopy Lab, Ariel University, Ariel 40700, Israel; tamarbr@ariel.ac.il; 5Department of Physics and Astronomy, Texas Christian University, Fort Worth, TX 76109, USA; lql3@cornell.edu (L.L.); dustin.johnson@tcu.edu (D.J.); y.strzhemechny@tcu.edu (Y.M.S.); 6Physics Department, Faculty of Natural Sciences, Ariel University, Ariel 40700, Israel

**Keywords:** nanophase, thin films, stabilization of the cubic phase of zirconia, magnetron co-sputtering

## Abstract

The cubic zirconia (ZrO_2_) is attractive for a broad range of applications. However, at room temperature, the cubic phase needs to be stabilized. The most studied stabilization method is the addition of the oxides of trivalent metals, such as Sc_2_O_3_. Another method is the stabilization of the cubic phase in nanostructures—nanopowders or nanocrystallites of pure zirconia. We studied the relationship between the size factor and the dopant concentration range for the formation and stabilization of the cubic phase in scandium-stabilized zirconia (ScSZ) films. The thin films of (ZrO_2_)_1−*x*_(Sc_2_O_3_)*_x_*, with *x* from 0 to 0.2, were deposited on room-temperature substrates by reactive direct current magnetron co-sputtering. The crystal structure of films with an average crystallite size of 85 Å was cubic at Sc_2_O_3_ content from 6.5 to 17.5 mol%, which is much broader than the range of 8–12 mol.% of the conventional deposition methods. The sputtering of ScSZ films on hot substrates resulted in a doubling of crystallite size and a decrease in the cubic phase range to 7.4–11 mol% of Sc_2_O_3_ content. This confirmed that the size of crystallites is one of the determining factors for expanding the concentration range for forming and stabilizing the cubic phase of ScSZ films.

## 1. Introduction

The crystal structure is one of the decisive factors determining the properties of materials [1]. To this point, zirconia (ZrO_2_) is a polymorphous material that exists in three crystallographic phases at atmospheric pressure: monoclinic stable up to 1205 °C, tetragonal from 1127 to 2396 °C, and cubic from 2369 to 2710 °C [2]. It was reported recently that the phase transition temperature depends on the film thickness [3]. The cubic zirconia demonstrates exceptional properties, such as high hardness [4] and dielectric constant [5], superior chemical stability [6], and prominent optical characteristics [7]. As a result, ZrO_2_ films have been exploited for a broad range of applications, e.g., wear-resistant coatings, medical implantology [8], oxygen detectors [9], thermal barrier coatings [10,11,12], solid electrolytes [13,14,15], and coatings for lenses and windows [16]. However, it is impossible to exploit pure zirconia in most of the above applications because of the high-temperature requirements of the c-phase existence. Thus, stabilizing the cubic zirconia at room temperature is critical for successfully applying this amazing material.

There are two strategies for the c-phase stabilization of zirconia at room temperature. The first is the most common concept related to introducing foreign impurities (doping) into ZrO_2_, proposed by Ruff et al. [17] in 1929. In the c-phase at high temperature, the Zr^4+^ cations occupy the sites of the fcc cell while the O^2−^ anions are arranged inside the tetrahedral voids. However, at temperatures below 2369 °C, the effective size of the Zr^4+^ cations is too small to provide big enough tetrahedral voids and sustain the cubic structure, so it has to be partially substituted with a larger cation of lower valence, e.g., by incorporating Y^3+^, Ca^2+^, or Mg^2+^ in the ZrO_2_ lattice. The second strategy of cubic zirconia stabilization does not involve any doping processes and, especially in the case of zirconia thin films, has been attributed to the size effect [18], energy input, and stress during the film growth [19,20], oxygen vacancies, and incorporation of N atoms in the ZrO_2_ lattice [21,22].

According to [2,22], the doping of zirconia by around and above 9 mol% of Sc_2_O_3_ leads to the c-phase of ZrO_2_ stabilization at room temperature and the formation of ScSZ. In ScSZ, the substitution of Zr^4+^ cations by Sc^3+^ results in one oxygen vacancy for each pair of scandium cations to maintain charge neutrality. Using the Kröger–Vink notation, this process is described by Expressions (1) and (2) as follows:(1)$$Sc_{2}O_{3}\overset{ZrO_{2}}{\to }2Sc_{Zr}^{\prime }+V_{\ddot{o}}+3O_{o}^{x}$$
(2)$$O_{o}^{x}\overset{\;}{↔}V_{\ddot{o}}+2e+\frac{1}{2}O_{2}$$

Here, $Sc_{Zr}^{\prime }$ is the Sc atom in the site of Zr with an apparent negative charge, $V_{\ddot{o}}$ is the double positively charged oxygen vacancy in the oxygen position, $O_{o}^{x}$ is the oxygen in a tetrahedral void with zero net charges, i.e., lattice oxygen. Note that, unlike the atoms of Y, Ca, or Mg, the ionic radii of Sc^3+^ and Zr^4+^ differ insignificantly, 87 and 84 pm, correspondingly [23], so the lattice distortion due to Sc doping is less significant than, for example, to Y. This is the critical factor responsible for the highest ionic conductivity of ScSZ among the other zirconia-based electrolytes [24,25,26]. However, conductivity strongly depends on temperature, and zirconia-based bicomponent electrolytes do not exhibit practical conductivity levels, except at temperatures of about 800 °C or higher [27]. Reducing electrolyte layer thickness from tens to a couple of microns, and even to hundreds of nanometers, is a prospective way to decrease the temperature of effective conductivity [28]. The control of phase stability and ionic conductivity of ScSZ may be achieved as well with the addition of a secondary and ternary co-dopant, such as ceria, bismuth oxide, yttria, ytterbia [29,30,31].

A wide range of both CVD and PVD methods, such as atomic layer deposition [32], plasma-enhanced chemical vapor deposition [33], spray pyrolysis [34], sol–gel [35], ink-jet printing [36], pulsed laser deposition [37], electron beam deposition [38], and sputtering [39] are used to deposit zirconium-based ceramics. Considering the unique properties and broad applicability of zirconium ceramics, it is unsurprising that many of those who review articles and monographs are devoted to studying these materials. However, most works are focused on studying commercially available powders and pellets of known composition or consider only a narrow concentration range. There are few works on the study of phase diagrams and the structure of bulk yttrium-stabilized zirconia (YSZ) and ScSZ over the entire range of mutual concentrations of Zr and dopant [2,39,40]. The authors revealed the c-phase of YSZ and ScSZ was stable at room temperature in a narrow concentration range, respectively, for YSZ at Y_2_O_3_ content from 6 to 10 mol.% and 8–12 mol% of Sc_2_O_3_ for ScSZ. According to the authors, the ScSZ phase compositions and transitions are more complicated than YSZ, and there are several variants of ScSZ phase diagrams for which phase limits vary. This phase diversity is due to the close ionic radii of Zr^4+^ and Sc^3+^ and, as a consequence, the absence of lattice stresses and the random redistribution of stabilizing impurity ions and oxygen vacancies. These facts, in turn, make the structure of ScSZ quite sensitive to the synthesis conditions and thermal history.

The authors [41] report forming the fcc phase of ScSZ monocrystalline thin films deposited by the Molecular Beam Epitaxy on a single crystal Al_2_O_3_ (0001) substrate with 4.6–17.6 mol% Sc_2_O_3_ content, which is considerably higher than 8–12 mol% of the solution or arc-melted processed ScSZ [2,39,40]. Thus, it reveals the possibility of forming c-phase ScSZ films in a much wider range of concentrations and confirms that the deposition methods, thermal history, and size factor are essential for the phase composition and stability of the ScSZ.

Plenty of works study the structure of YSZ and ScSZ thin films, but again, only for specific concentrations of dopants [38,42]. The authors of [1,13,43] demonstrate that the film thickness, deposition, and annealing conditions are the key factors determining its structure, morphology, and, consequently, physical properties. In this regard, magnetron co-sputtering is an effective method for depositing bi- and three-component ZrO_2_-based thin films in a wide range of mutual concentrations for a thorough study of their structure and properties.

Presently, for developing highly efficient and stable solid oxide fuel cells (SOFC), the magnetron sputtering method is used to synthesize compact solid electrolytes in several laboratories [2,13,15,44]. It should be noted that mainly YSZ electrolytes are synthesized; in most cases, YZr composite metal targets are used in the sputtering process, the Y content of which varies from 15 to 20 at%. The reactive sputtering of these targets in an (Ar)_0.8_(O_2_)_0.2_ atmosphere leads to the formation of compact polycrystalline (Y_2_O_3_)*_x_*(ZrO_2_)_1−*x*_ films of the fcc structure [42]. Note that the targets’ constant concentration imposes restrictions on the ability to tune the content of the components in the films and, thus, change their physical properties in a highly precise manner. Applying joint magnetron sputtering (magnetron co-sputtering, MC) removes these limitations, providing another tool for precision-tuning the characteristics of the deposited films. The essence of the MC method lies in the simultaneous sputtering of targets from two or more magnetron sources—“guns”. A change in the input power makes it possible to vary and control the concentration composition of the deposited films over a wide range.

This study reveals the c-phase of ScSZ thin film formation in a broad concentration range (6.5 to 17.5 mol%) of Sc_2_O_3_ deposited by reactive DC magnetron co-sputtering of Sc and Zr targets on glass and quartz non-preheated substrates. The c-phase ScSZ stability on temperature treatment is critical for applying these films as solid electrolytes in SOFC [45]. Therefore, the influence of subsequent annealing at 550 and 1100 °C (low-temperature SOFCs operating temperature and maximum sintering temperature of SOFC functional layers, respectively) in the air on the structure, lattice parameters, and morphology of ScSZ sputtered thin films was also studied. The ScsZ films magnetron sputtering on the substrates heated at 450 °C led to shrinking the c-phase interval to the 7.4–11 mol% of Sc_2_O_3_ content. Comparative X-ray analysis revealed the determining influence of crystallite sizes (nanocrystallinity of the film) on the concentration range of the cubic phase formation.

## 2. Methods and Materials

### 2.1. Reactive Magnetron Co-Sputtering of ScSZ

Reactive co-sputtering of ScSZ films was carried out in a VST TSFP-842 balanced vacuum magnetron sputtering system (VST, Petah Tikva, Israel) in DC mode. The films were deposited on Soda Lime Glass 0215 (Corning Glass, Corning, NY, USA) and quartz (SPI Supplies, West Chester, PA, USA) substrates with the size of 25 mm × 25 mm × 1 mm, precleaned according to the protocol presented in the Appendix A. Before sputtering, the VST chamber was evacuated to a pressure of 3·10^−6^ Torr. The two targets, Sc (99.99%) (American Elements, Los Angeles, CA, USA) 1/8-inch thickness and Zr (99.9%) (Testbourne, Whitchurch, UK) 1/8-inch thickness and 2-inch in diameter, were co-sputtered.

The pressure of the gas mixture in the VST chamber was 7 mTorr; Ar was used as the working reference gas, and O_2_ was the reactive one. The composition of the gas mixture was determined and controlled by digital mass flow controllers MKS GE50A (MKS, Andover, MA, USA) separately for oxygen and argon gas lines. The reactive gas flow was kept constant at 1.4 sccm. The argon flow was automatically varied from 4.4 to 4.6 sccm by a digital sputter tuning system to stabilize the plasma discharge and the specified working pressure in the chamber. Thus, the composition of the working gas mixture varied from 23 to 24 vol% of oxygen in argon. The samples were deposited from the so-called “early oxide” mode. In this regime, argon ions almost wholly knock out the oxide layer formed on the target surface, so the target is not “overgrown” with oxides, making it possible to maintain a relatively high level of secondary electron emission and sputtering efficiency. In this case, the final oxidation of the sputtered particles occurs on the substrate surface during the film growth.

It should be noted that the realization of such a sputtering mode required the modernization of the lines that supply the working and reaction gas. Concentrating most oxygen near the substrate area while evenly distributing argon over the chamber was necessary. Therefore, a ring-shaped oxygen pipeline was mounted directly above the sample table. At the same time, argon was supplied through two gas pipelines on both sides of the chamber opening (see Appendix A).

The DC power applied on the Sc target varied from 0 to 160 W, depending on the goal composition of the ScSZ films, while the power on the Zr target did not change and was kept equal to 150 W DC. The distance between the substrate surface and the targets was 80 mm. The stage with substrates was rotated at 10 rpm. The ScSZ films were sputtered under two temperature conditions, the first without intentional heating of the sample stage (RT–room temperature) during the deposition and the second at 450 °C of the sample stage. The duration of the RT co-sputtering for all the samples was one hour, and 45 min for all samples deposited at 450 °C.

The ScSZ films, co-sputtered on the quartz substrates at RT, were annealed at 550 and 1100 °C for 120 min in the Tube Furnace KJ-T1200 (Zhengzhou Kejia Furnace Co., Ltd., Zhengzhou, China) in an ambient air environment. The annealing thermograms are presented in Appendix A.

### 2.2. Sample Characterization

The structural characteristics of the co-sputtered ScSZ films were determined with an X-ray powder diffractometer (SmartLab SE, Rigaku, Tokyo, Japan) with Cu-Kα radiation (λ = 1.5460 Å). The XRD patterns were registered in 2θ geometry (angle of incidence 3°) in the range of 20–80° with a step of 0.03° and a rate of 0.5°/min. The phase analysis was performed using SmartLab Studio II ver. 4.2.44.0, using the Rietveld Refinement method in the 20–70° range. The crystallite size and lattice parameter were calculated using the Comprehensive Analysis module and the Halder–Wagner modeling method. The films’ thickness, surface morphology, and chemical composition were revealed with scanning electron microscopy (SEM) MAIA3 (TESCAN, Brno, Czech Republic) equipped with an X-ray energy dispersive spectrometer (EDS) X-Max^N^ (Oxford Instruments, Abingdon, UK). Auger Electron Spectroscopy (AES) EG3000, CMA2000 (LK Technologies, Maple Heights, OH, USA) was applied in conjunction with argon remote plasma treatment to estimate the surface stoichiometry of the ScSZ films. The specific masses of the deposited films were evaluated via the gravimetric method [46] with the analytical balance BM-5 (A&D Company Limited, Tokyo, Japan).

## 3. Results and Discussion

### 3.1. Concentration Analysis

The ScSz films were deposited on glass and quartz substrates by the reactive DC magnetron co-sputtering of both Zr and Sc targets at RT and 450 °C temperatures. The power on the Zr target was kept constant (150 W), while the power on the Sc varied from 0 to 160 W. Sc and Zr’s content in deposited films was revealed with EDS and AES (see Appendix A) at atomic % and then was recalculated to the molecular concentration of Sc_2_O_3_ in ZrO_2_ with Formula (3):(3)$$Mol_{Sc_{2}O_{3}}=\left(\left(\frac{At_{Sc}}{At_{Sc}+At_{Zr}}\right)/\left(2-\left(\frac{At_{Sc}}{At_{Sc}+At_{Zr}}\right)\right)\right)\times 100\%$$
here, *At_Sc_* and *At_Zr_* are the EDS-revealed scandium and zirconium atomic concentrations, respectively. The concentration dependences of Sc_2_O_3_ in the RT-sputtered ScSZ films on the power applied to the Sc target revealed with EDS and AES are presented in Figure 1.

Linear approximation of the data presented in Figure 1 allows for determining the coefficient *k* = 0.125 to calculate the power we need to apply to the Sc target to deposit films of the goal concentrations at the RT temperature regime. Particular attention was paid to the interval from 5 to 13 mol% Sc_2_O_3_ because of ScSZ c-phase expectation here. Therefore, the power applied to the Sc target was 5 W in this concentration range. Moreover, the AES studies were done in this concentration interval to check the regularity of Sc distribution over the film’s surface and thickness. The high-level match of the EDS and AES data points to the concentration uniformity of the deposited ScSZ films over the volume.

The concentration dependence of Sc_2_O_3_ in the co-sputtered at 450 °C ScSZ films on the power applied to the Sc target revealed with EDS is presented in Figure 2.

The concentration dependence shown in Figure 2 is linear also, but its slope is 35% less than in RT deposition. An increase in the substrate temperature has a twofold effect on the film growth. On the one hand, it reduces the deposition rate due to a rise in the energy of the adatoms, thus increasing their diffusion pass along the substrate surface. On the other hand, the heat irradiated by the substrate also increases the temperature of the targets; hence, raising the sputtering yield, which, in turn, leads to an increase in the deposition rate. Therefore, the “hot” deposition time was adjusted using the gravimetrical method to deposit the ScSZ films with almost the same specific masses as in the case of the ScSZ sputtered at RT (on the non-preheated substrates). The corresponding deposition time was 45 min.

### 3.2. Crystal Structure, Crystallite Sizes, and Lattice Parameters

To determine the concentration region of the fcc phase and to analyze the effect of annealing at temperatures of 550 and 1100 °C on the structural characteristics, X-ray diffraction studies of annealed and as-deposited ScSZ films were carried out. The XRD patterns of as-deposited at RT and annealed at 550 and 1100 °C ScSZ films with different content of Sc_2_O_3_ are presented in Figure 3a,b,c, respectively. The phase analysis of the diffraction patterns was performed using reference DB cards: 01-070-6633; 01-083-0940 and 01-077-0724 for cubic, monoclinic and trigonal phases of the ScSZ films correspondingly. The respective peaks of the diffraction patterns are indexed. All films for XRD studies were deposited on quartz substrates.

The deposited ScSZ films were polycrystalline, uniform, and transparent without cracks and defects. Comprehensive phase analysis with “Powder XRD ver. 4.0” software of XRD patterns of as-deposited ScSZ films with 1.25 mol% Sc_2_O_3_ concentration (see the black curve in Figure 3a) revealed the reflections of the monoclinic (P121/c1) phase. The halo detected in the low-angle region corresponds to the ultradisperse or amorphous phases of ScSZ film and diffuse scattering from the quartz substrate. The intensity of the halo decreases with increasing Sc_2_O_3_ content, and then, at 18.75 mol% and higher, it becomes more pronounced (see Figure 3a) again. The halo decreases and partially transforms into crystal modification after annealing at 550 °C. After treating the films with Sc_2_O_3_ content of more than 7% at 1100 °C, the only halo from the quartz substrate remains. It is important to note that, at concentrations of 3.75 mol% and less, the halo remains pronounced even after annealing at 1100 °C (black and red curves in Figure 3c). These observations indicate that the halos from ScSZ films with Sc_2_O_3_ content above and below 3.75 mol% are different. In the first case, the halo predominantly consists of ultrafine crystallites, so-called subcritical nuclei of the crystalline phase, and a small amount of amorphous, “translational glass” phase. During the annealing, the ultradisperse crystallites merge and form the fcc phase, thus contributing to the corresponding reflections, and the halo disappears. At 3.75 mol% Sc_2_O_3_ content and below, even after annealing at 1100 °C, the halo remains pronounced and intense (see Figure 3c), indicating the amorphous state of the phase forming it. Such behaviors are characteristic of glasses with subcritical nuclei of crystalline phases and without them (correspondingly up and below 3.75 mol% of Sc_2_O_3_) upon treatment near phase transition temperatures [47,48,49]. Recall that the monoclinic-to-tetragonal phase transition temperature for ScSZ is 1205 °C at atmospheric pressure [2]. The conventional XRD methods are unable to reveal subcritical nuclei and distinguish these amorphous and ultradisperse phases. Special electron diffraction techniques such as THEED (transmission high-energy electron diffraction) and FTEM (fluctuation transmission electron microscopy) are needed [50,51,52].

The Rietveld analysis of the diffraction patterns revealed the crystal structure of the as-deposited ScSZ films with Sc_2_O_3_ content from 6.5 to 17.5 mol%, which is described by the Fm3m space group of the cubic syngony. The intensity of the diffraction peaks increases, and the half-width decreases with an increase in the annealing temperature. After treatment at 1100 °C, the ScSZ films of the c-phase are characterized by pronounced texture along the ˂200˃ direction. An analysis of the intensity distribution and position of the diffraction maxima of X-ray patterns from ScSZ films with Sc_2_O_3_ content of 18.75 mol% indicates the formation of a highly textured, along the ˂21-2˃ rhombohedral (R-3), phase after annealing at 1100 °C.

Studying the concentration dependence of the lattice parameters (LP) and the sizes of the coherent scattering region (CSR) revealed from XRD data can provide additional information about the film structure and its formation processes.

Let us consider the concentration dependence of the LP and CSR of as-deposited ScSZ films at Sc_2_O_3_ content less than 8 mol% in the framework of the semi-empirical phenomenological models of solid solutions (SS) [53]. The SS’s regularity (randomness of the solution’s components distribution) is one of the fundamental concepts of solid-state physics. The separate sputtering of Zr and Sc targets without reactive gas (O_2_) revealed almost the same specific growth coefficients of Zr and Sc films equal to 0.190 and 0.176 nm/(W × min), respectively (see Appendix A). These results are consistent with the sputtering yield of Sc and Zr atoms by Ar ions with an energy of 500 eV at a temperature of 0 °C (0.634 and 0.557, respectively) calculated using a semi-empirical model [54]. Taking into account the physical characteristics of Sc and Zr, according to the Hume–Rothery criteria [55], these metals form a continuous series of regular solid solutions over the entire concentration range [56]. Let us pay attention to the essential details of our experiment. The ScSZ film deposition was carried out by the co-sputtering of Sc and Zr metal targets with a deficiency of the reactive gas (O_2_) supplied directly above the substrate surface. Thus, considering the high energy of the sputtering process, the atoms of the Sc and Zr metals were knocked out of the corresponding targets, and the predominant oxidation of the metallic atoms occurred on the substrate surface during the film formation without an advantage in adsorption for both Zr and Sc atoms. So, the assumption of the random distribution of Sc and Zr atoms in the solid alloy has a substantial basis.

The concentration dependence of the fcc LP and CSR size of as-deposited ScSZ films revealed with full-profile analysis of the corresponding XRD patterns are presented in Figure 4.

According to X-ray analysis, the ScSZ films in the grey, orange, and blue areas presented in Figure 3 are not single-phase. Therefore, note that the only fcc-phase LP and CSR concentration dependence is shown in Figure 3. Up to 3.5 mol% Sc_2_O_3_ content (see the grey region in Figure 4), the ScSZ films are characterized by the monoclinic (P121/c1) phase with an intense halo. Based on the regularity of the ScSZ sold alloy, at low concentrations, single scandium atoms (having neighbors in the first coordination sphere of only zirconium atoms) dominate (Appendix A) by analogy with the cluster approach developed by the authors [57,58,59] for close-packed solids. This does not favor the formation of fcc phase regions. So, up to 3.5 mol% Sc_2_O_3_ content, the monoclinic and amorphous phases of ScSZ prevail. At concentrations above 3.5 mol%, the fcc diffraction maxima appear along with monoclinic reflections and diffuse halo and become dominant at 5 mol%. The monoclinic phase reflections fade away at 6.5 mol% of Sc_2_O_3_, but the halo is still pronounced, although less intense. Note that according to the cluster approach, the number of single Sc atoms (single cluster) decreases with an increasing Sc_2_O_3_ content above 4 mol%. In contrast, the concentration of effective triple Sc clusters starts to rise rapidly (Appendix A). This leads to increasing the fcc phase nuclei number, consequently decreasing the CSR size and increasing the lattice parameter, which is observed in Figure 4 in the concentration range from 3.5 to 7.5 mol % of Sc_2_O_3_. With a subsequent increase in concentration, the number of single Sc clusters fades away, while the number of “effective triple” Sc clusters continues to grow and become dominant. These effective triples are, in fact, the nuclei of the fcc phase in ScSZ. Therefore, fcc phase nuclei are predominantly formed on the substrate, the subsequent coalescence of which leads to large CSRs and an abrupt decrease in the lattice parameter observed in Figure 4 at 8.75 mol%. We also note that the Sc concentration growth occurs due to increased power supplied to the Sc target. It results in an increase in the energy of adatoms in the diffuse layer and the substrate temperature, which also contributes to the coalescence of nuclei and CSR growth. From 8.75 to 15 mol% of Sc_2_O_3,_ the behavior of the LP of as-deposited ScSZ films correlates with CSR. Decreasing the sizes of CSR leads to an increase in LP. Let us pay attention to the LP shelf at 15 to 18 mol% of Sc_2_O_3_ content and a corresponding decrease in CSR size of 6%. According to Vegard’s law [60], the absence of the LP response to a change in concentration is one of the criteria of the SS decomposition process [58]. Facing the insignificant difference in the ionic radii of Sc^3+^ and Zr^4+^, 87 and 84 pm, correspondingly [23], continuous changes in the number of oxygen vacancies, and the presence of only fcc reflections on the XRD patterns, the statement about the phase heterogeneity is not indisputable. On the contrary, the appearance of the low-angle halo, along with the decrease in the fcc peak intensity (see Figure 3a) above 18 mol% Sc_2_O_3_ content, definitely indicates multiphase film formation.

The XRD analysis of the ScSZ films annealed at 550 and 1100 °C confirmed the multiphase composition at concentrations above 18 mol% of Sc_2_O_3_. The concentration dependence of the fcc LP and CSR size of ScSZ films annealed at 550 and 1100 °C are presented in Figs. 5 and 6, respectively. The annealing of the ScSZ films at 550 °C led to a decrease in the LP from 5.125 ± 0.005 to 5.069 ± 0.003 Å, but did not affect the CSR size at a low concentration range (till 3.75 mol% Sc_2_O_3_). Let us pay attention to the increase in CSR from 3.75 to 8 mol% of Sc_2_O_3_, associated with coalescence during the annealing of subcritical nuclei of the fcc phase. This fusion is evidenced by the decrease in the halo intensity and the LP jump leveling at 8 mol% content. The tendency for the LP to decrease with the increase in concentration (see Figure 5) can be facilitated by the rise in oxygen vacancies. Let us pay attention to the LP minimum and CSR maximum in Figure 5 in the vicinity of 11.25 mol% Sc_2_O_3_. The authors [61,62,63] show that the ScSZ is characterized by maximum conductivity at about 11 mol% Sc_2_O_3_ concentration range. Note that annealing at 550 °C did not lead to the disappearance of the small-angle halo or the emergence of new diffraction reflections in the concentration range 18–22 mol% (see Figure 3b).

Annealing the ScSZ films at 1100 °C (see Figure 6) led to a decrease in the LP, associated with the coarsening of the film grains and an almost twofold increase in the CSR. The reflections corresponding to the ScSZ rhombohedral phase (R-3) were recorded in the diffraction patterns from the films with a scandium content above 18 mol%. In this case, the low-angle halo (purple X-ray diffraction pattern in Figure 3c) disappeared, and the nanocrystallites that formed it became the nuclei of the rhombohedral phase, thus initiating the Fm3m-R3 transition. 

We focus on the fact that the concentration interval for the c-phase ScSZ film formation, according to the results of the analysis of X-ray data presented in this work, is more than two and a half times wider than in the case of the ScSZ synthesis by conventional methods [2,34,35,40]. The authors [18,64,65] report that a decrease in the size of crystallites below 15 nm contributes to the formation and stabilization of the cubic phase of zirconia, even without doping. Taking into account that, according to the data of X-ray analysis, the crystallite sizes of the as-deposited RT films did not exceed 85 Å (see Figure 4), we assumed that it was the size factor that contributed to such a significant expansion of the interval of the c-phase of the ScSZ films. The deposition on a “hot” substrate is an effective way to increase crystallite sizes [66]. So, to verify the “size” suggestion, the magnetron DC co-spattering of the same Zr and Sc targets at the same experimental conditions was conducted on quartz substrates heated and stabilized at 450 °C. The XRD patterns of the SCSZ films deposited on quartz substrates at 450 °C are presented in Figure 7. 

The monoclinic and cubic phase reflections were revealed on the diffraction pattern from the ScSZ films with Sc_2_O_3_ content of 5 mol% (Figure 7, black line). At concentrations from 7.4 to 11 mol%, the ScSZ films are single-phase and are characterized by the Fm3m symmetry space group (Figure 7, red line). When the Sc_2_O_3_ content is 11.5 mol%, weak peaks of the trigonal phase R-3c appear in the diffraction pattern, along with reflections from the cubic phase. Thus, based on X-ray analysis data, it has been established that when spattering onto substrates heated to 450 °C, the c-phase ScSZ films are formed in the range from 7.4 to 11 mol% Sc_2_O_3_ content.

The Rietveld Refinement and the Halder–Wagner modeling were applied to the XRD patterns to reveal the lattice parameter and crystallite size dependences on Sc_2_O_3_ content. Note that Figure 8 presents data for only the c-phase ScSZ.

Let us note that, in the Sc_2_O_3_ concentration range from 7.4 to 11 mol%, the ScSZ crystallite size does not exceed 11 nm, and, precisely in this range, the single-phase films of the cubic structure are formed. Less symmetrical phases, along with cubic ones, are present in neighboring concentration ranges. In these regions, the sizes of the crystallites of the c-phase are larger. Notably, at low Sc_2_O_3_ concentrations, the crystallites of the cubic phase are twice as large. The presented results are consistent with [18,64,65]. The authors report that a decrease in the size of crystallites below 15 nm contributes to the formation and stabilization of the c-phase zirconia. Decreasing the c-phase range while increasing the crystallite size confirms the suggestion that the size of crystallites is one of the determining factors for expanding the concentration range of forming and stabilizing the cubic phase of ScSZ films.

### 3.3. Morphology of the As-Deposited and Annealed RT Co-Sputtered ScSZ Films

The dependence of the c-phase ScSZ film morphology on the annealing temperatures and Sc_2_O_3_ concentration is shown in the SEM images in Figure 9. The morphologies of the RT as-deposited 6.25 and 17.5 mol% ScSZ are very similar (see Figure 9(A1,A3)). The tiny (~5–8 nm CSR, see Figure 4) crystallites combine into columnar-shaped (~50 nm in diameter) grains, forming the films with 170 and 150 nm thickness for 6.25 and 17.5 mol% ScSZ, respectively. The corresponding cross-section SEM images of RT as-deposited ScSZ films are presented in Figure 10. Note the more distinct grain boundaries of 17.5 mol% ScSZ films and sharper size distribution than the 6.25 mol% one. The grains of the 10 mol% ScSZ are less pronounced and uniform (see Figure 9(A2)). The film annealing at 550 °C had no significant effect on the grain size of both 6.25 and 17.5 mol% ScSZ, but the grain boundaries of 17.5 mol% became sharper. 

The annealing at 550 °C of 10 mol% ScSZ leads to the merging of small grains and the formation of globular-shaped morphology. According to the XRD data, the 6.25, 10, and 17.5 mol% ScSZ films annealing at 550 °C increased the CSR by 18, 36, and 30%, respectively. Increasing the annealing temperature to 1100 °C led to a threefold increase in CSR (Figure 6) and the formation of large grains with distinct boundaries (see Figure 9C). Again, the morphologies of 6.25 and 17.5 mol% ScSZ films are very similar. In comparison, the grains of 10 mol% ScSZ are bigger with smooth and somewhat merged boundaries (see Figure 9(C2)). The cross-section SEM images of RT as-deposited ScSZ films presented in Figure 10 confirm their columnar structure.

The 100 nm gold was sputtered on top of the ScSZ before the cross-section was taken to enhance the image contrast and suppress the charge effect. Recall that the deposition time for all RT films was 1 h. To increase the Sc_2_O_3_ concentration, the power applied to the scandium target was increased, leading to a rise in the specific mass of the deposited film. So, the nonlinear behavior of the film’s thickness with doping content, revealed from the cross-sections, becomes entirely unexpected. The ScSZ film’s specific mass on the Sc_2_O_3_ content is presented in Figure 11a, and their linearity, together with unexpected thickness behavior, indicates the nonlinear dependence of the density of the as-deposited films shown in Figure 11b on the impurity content.

An increase in the Sc_2_O_3_ content to 6.25 mol% decreases the film’s density from 4.8 to 3.6 g/cm^3^. Further increase in concentration leads to a rise in ScSZ density that reaches a maximum of 5.6 g/cm^3^ at 12.5 mol%, which is close to the theoretical estimation of 6.06 g/cm^3^ for zirconia [64]. Recall that the lattice parameter and CSR of the ScSZ films deposited at RT also have singularities in the vicinity of this concentration. An increase in Sc_2_O_3_ content above 18 mol% leads to a sharp decrease in density to 3.7 g/cm^3^ at 21.25 mol%. Note that the behavior of the ScSZ density dependence on Sc_2_O_3_ content correlates well with the XRD and SEM data.

The cross-section SEM images of the RT as-deposited and annealed at 550 and 1100 °C ScSZ films with impurity content of 12.5 and 17.5 mol% are presented in Figure 12a,b, respectively.

The annealing at 1100 °C of the as-deposited 12.5 mol% ScSZ caused the film’s thickness to decrease by only 6% (see Figure 9A), while the annealing of 17.5 mol% Sc_2_O_3_ content one—by 14% (see Figure 9B). This result confirms the above conclusion regarding the higher density of the 12.5 mol% ScSZ film than the 17.5 mol% one. Moreover, this fact indicates that temperature exposure at 1100 °C not only does not destroy RT-deposited ScSZ films with a Sc_2_O_3_ content of 12.5% but also improves their density and crystallinity.

## 4. Conclusions

The (ZrO_2_)_1−x_(Sc_2_O_3_)_x_ thin films with x from 0 to 0.2 were deposited on glass and quartz substrates at RT (room temperature) and 450 °C by reactive DC magnetron co-sputtering. The XRD analysis revealed the transformation of the RT films’ phase composition starting from monoclinic (P121/c1) at low concentrations to cubic (Fm3m) from 6.5 to 17.5 mol% of Sc_2_O_3_ and finally to rhombohedral (R-3) modification. The 8–12 mol% range of Sc_2_O_3_ content characteristic for c-phase ScSZ deposited by arc-melting methods was increased to 6.5–17.5 mol% of Sc_2_O_3_ in the case of reactive DC co-sputtered ScSZ thin films at RT. The combined effect of the size factor, the energy of the sputtered atoms, doping, oxidation, and growth processes caused an almost threefold expansion of the ScSZ c-phase formation range. The reactive DC co-sputtering of the Sc and Zr targets on the substrates at 450 °C led to the formation of the c-phase ScSZ only in the concentration range from 7.4 to 11 mol% of Sc_2_O_3_ with two times bigger CSR. This confirms and emphasizes the determining role of the crystallite size factor in the formation of the ScSZ c-phase films.

Annealing the RT as-deposited c-phase ScSZ films in the air at 550 and 1100 °C did not lead to phase transitions or structural damage. Moreover, such temperature treatment decreases the fcc lattice parameter and increases crystallite and grain size. It was found that ScSZ films with 12.5 mol% Sc_2_O_3_ had the highest density based on cross-SEM and gravimetric analyses.

## Figures and Tables

**Figure 1 nanomaterials-14-00708-f001:**
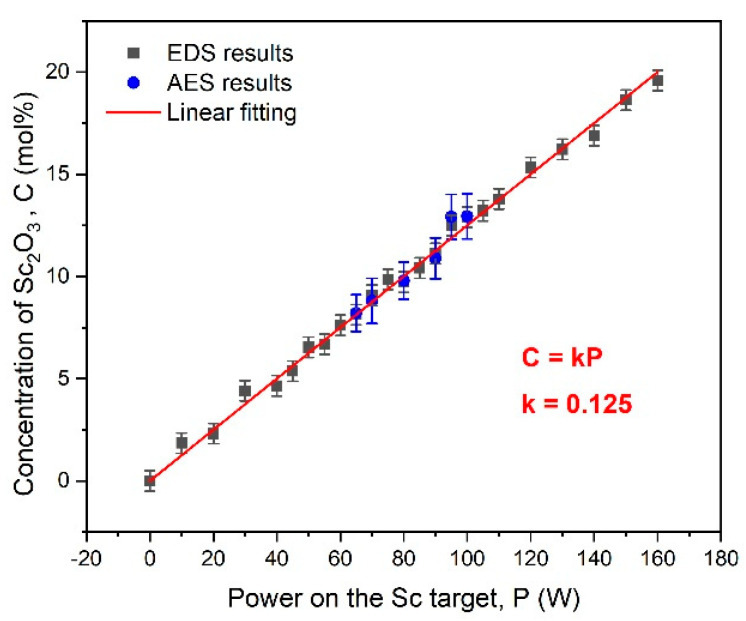
The concentration of Sc_2_O_3_ in ScSZ films co-sputtered at RT on power applied to the Sc target. Black squares are the results of the EDS study, and the blue circles are the AES data. A linear fitting of EDS and AES data is presented with the red line.

**Figure 2 nanomaterials-14-00708-f002:**
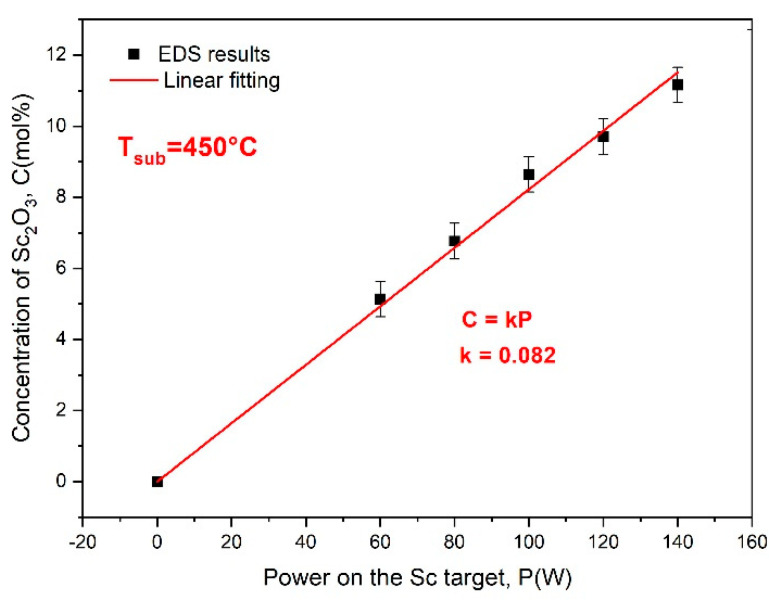
The concentration of Sc_2_O_3_ in the ScSZ films, co-sputtered at 450 °C, on the power applied to the Sc target. Black squares are the results of the EDS study. The red line shows a linear fitting of EDS data.

**Figure 3 nanomaterials-14-00708-f003:**
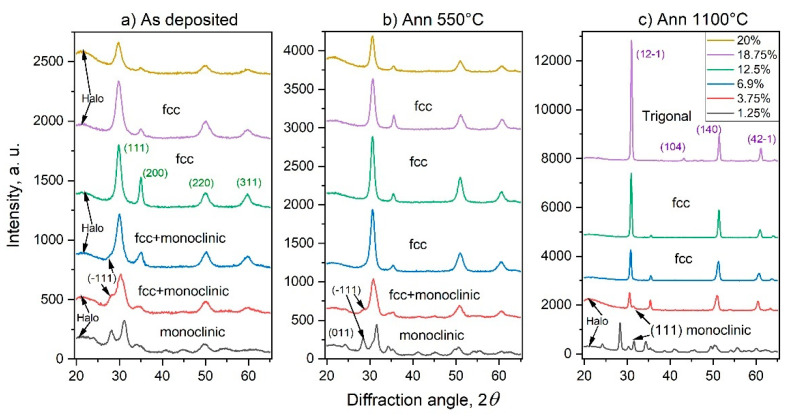
The diffraction patterns of (**a**) as-deposited at RT and annealed at (**b**) 550 and (**c**) 1100 °C ScSZ films. The content of Sc_2_O_3_ in the ScSZ films and annealing regimes are indicated in the figure.

**Figure 4 nanomaterials-14-00708-f004:**
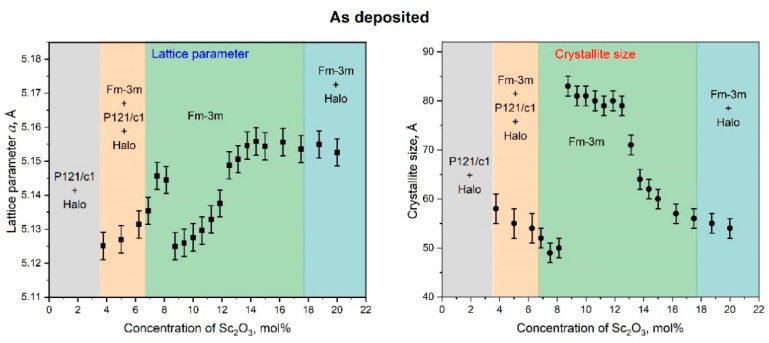
The dependence of lattice parameter and crystallite size of as-deposited ScSZ films on the Sc_2_O_3_ content. Note that only fcc phase lattice parameters are shown.

**Figure 5 nanomaterials-14-00708-f005:**
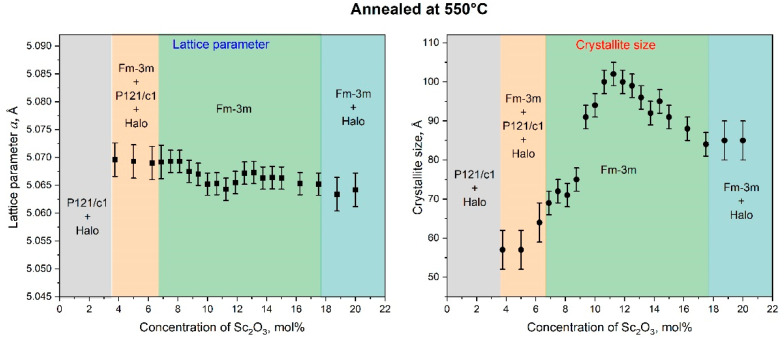
The dependence of the lattice parameter and crystallite size of ScSZ films annealed at 550 °C on the Sc_2_O_3_ content. Note that only fcc phase lattice parameters are shown.

**Figure 6 nanomaterials-14-00708-f006:**
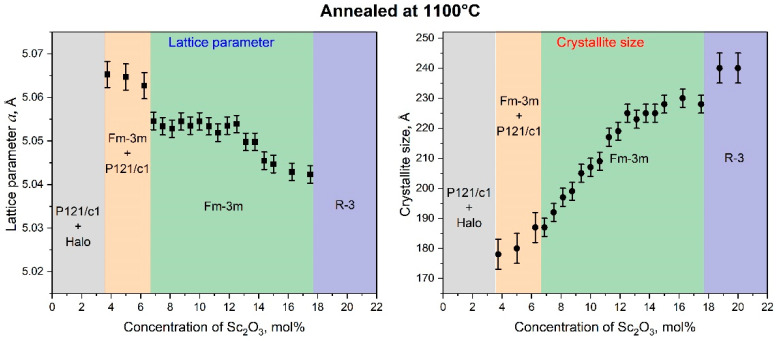
The dependence of the lattice parameter and crystallite size of ScSZ films annealed at 1100 °C on the Sc_2_O_3_ content. Note that only fcc phase lattice parameters are shown.

**Figure 7 nanomaterials-14-00708-f007:**
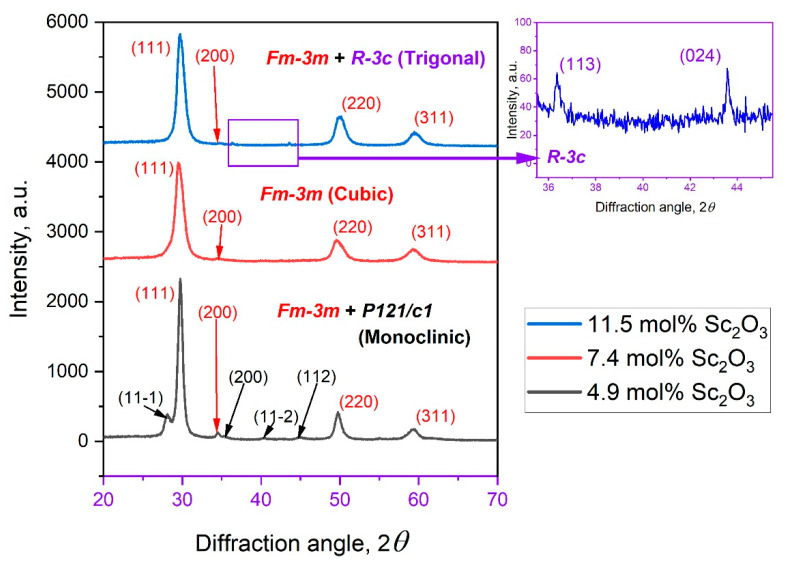
The diffraction patterns of the ScSZ films co-sputtered on quartz substrates at 450 °C.

**Figure 8 nanomaterials-14-00708-f008:**
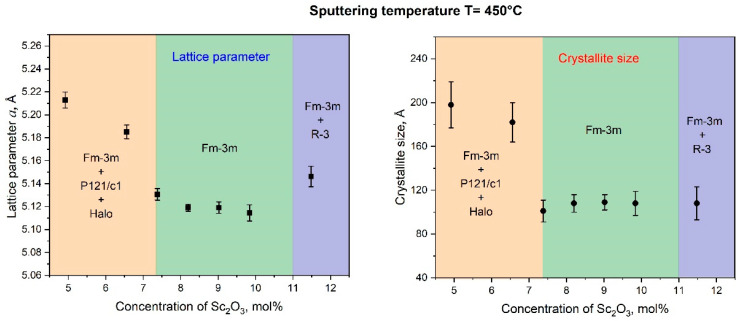
The lattice parameter and crystalline size dependence of c-phase ScSZ on Sc_2_O_3_ content.

**Figure 9 nanomaterials-14-00708-f009:**
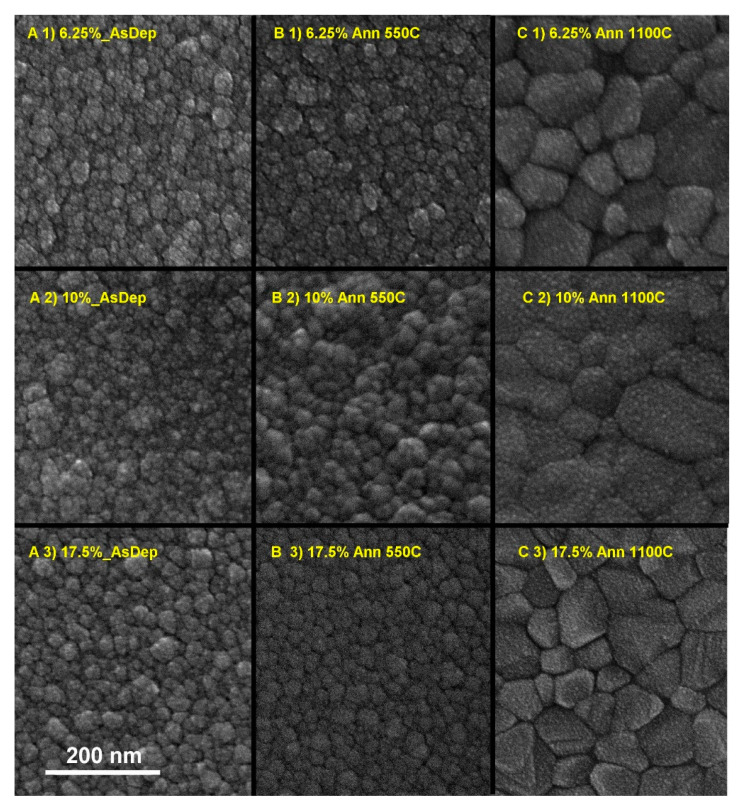
The SEM images of ScSZ films with 6.25, 10, and 17.5 mol% of Sc_2_O_3_—(**1**), (**2**), and (**3**) rows, respectively. Columns (**A**,**B**,**C**) correspond to temperature treatment conditions: as-deposited, annealed at 550, and 1100 °C, respectively.

**Figure 10 nanomaterials-14-00708-f010:**
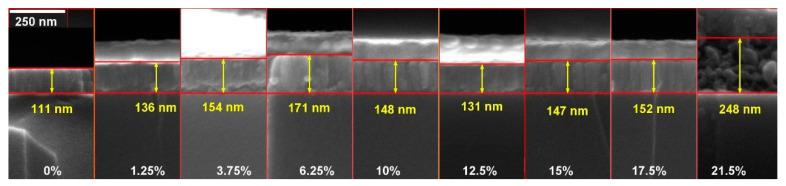
The cross-section SEM images of as-deposited ScSZ films. The gold film was evaporated on top of the sample’s surface to suppress the charge effect.

**Figure 11 nanomaterials-14-00708-f011:**
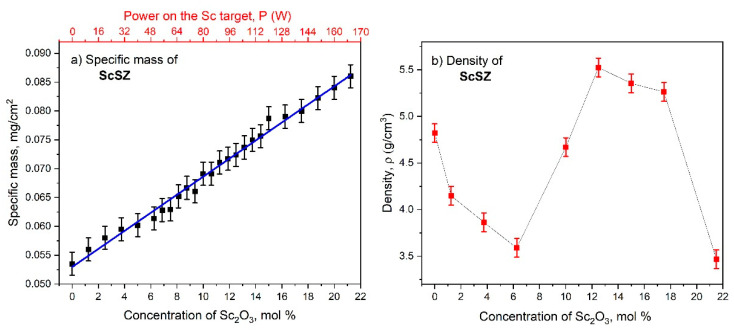
The dependencies of the (**a**) specific mass and (**b**) density of the RT as-deposited ScSZ films on Sc_2_O_3_ concentration.

**Figure 12 nanomaterials-14-00708-f012:**
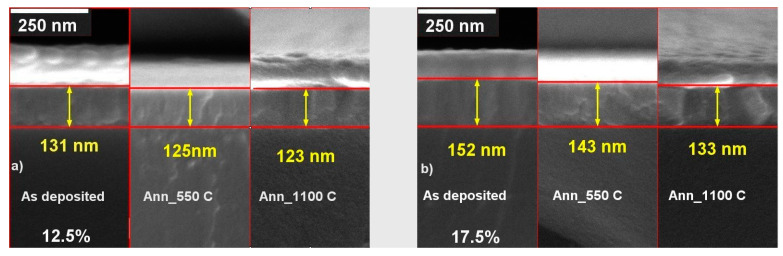
The cross-section SEM images of the RT deposited ScSZ films with (**a**) 12.5 and (**b**) 17.5 mol% of Sc_2_O_3_ content as-deposited and annealed at 550 and 1100 °C. The gold film was evaporated on top of the sample’s surface to suppress the charge effect.

## Data Availability

Data are contained within the article.

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
