# Peer review of "Nanocrystalline Cubic Phase Scandium-Stabilized Zirconia Thin Films"

_nanomaterials, 2024, doi:10.3390/nano14080708_

Round 1

Reviewer 1 Report

Comments and Suggestions for Authors

In this MS, the authors reported the relationship between the size factor and the dopant concentration range for the formation and stabilization of the cubic phase in scandium-stabilized zirco-nia (ScSZ) films. The results show that the size of crystallites is one of the determining factors for expanding the concentration range for forming and stabilizing the cubic phase of ScSZ films. I recommend publishing this manuscript. However, there are still some issues that need to be addressed before recommending publication.

1. In the XRD patterns, the standard PDF card of cubic or monoclinic phase ZrO2 should be presented in the picture.

2. The font of the paper is not standardized, and not uniform.

3. References from the last three years need to be cited.

Comments on the Quality of English Language

None

Author Response

We would like to thank the reviewer for the constructive and valuable comments, as well as for your time and consideration. We hope that the changes we have made will improve our article.

Reviewer 1

Comments and Suggestions for Authors

In this MS, the authors reported the relationship between the size factor and the dopant concentration range for the formation and stabilization of the cubic phase in scandium-stabilized zirconia (ScSZ) films. The results show that the size of crystallites is one of the determining factors for expanding the concentration range to form and stabilize the cubic phase of ScSZ films. I recommend publishing this manuscript. However, there are still some issues that need to be addressed before recommending publication.

  1. In the XRD patterns, the standard PDF card of cubic or monoclinic phase ZrO2 should be presented in the picture.

Thank you for your comment. However, in Figure 3 each of the 3 groups contains more than 5 XRD patterns . If we add to them 3 more standard patterns for different phases, the figure may seem overloaded and unclear. Instead, peaks corresponding to different planes of monoclinic and cubic ZrO2 are marked at patterns.  The sentence: “The phase analysis of the diffraction patterns was performed using reference DB cards: 01-070-6633; 01-083-0940 and 01-077-0724 for cubic, monoclinic and trigonal phases of the ScSZ films correspondingly. The respective peaks of the presented in Fig. 3 and Fig. 7 diffraction patterns are indexed.”  was added to the manuscript in Section 3.2, page 9.

  1. The font of the paper is not standardized, and not uniform.

Thank you for carefully checking our manuscript, we have corrected the font differences we found. In preparation for publication, further formatting of the manuscript will be done according to the template.

  1. References from the last three years need to be cited.

This is an important remark. New references were added to the Introduction; they are highlighted in yellow; see Introduction, pages 1 and 3.

Comments on the Quality of English Language

None

Submission Date

14 March 2024

Date of this review

02 Apr 2024 16:37:20

Reviewer 2 Report

Comments and Suggestions for Authors

Review of nanomaterials-2940208, entitled ‘Nanocrystalline cubic phase scandium-stabilized zirconia thin films’

The current manuscript focuses on the size factor and the dopant concentration range for the formation and stabilization of the cubic phase in scandium-stabilized zirconia (ScSZ) films. It is well-organized and written, it would be great helpful for understanding the stabilization of ScSZ. It is recommended to be accepted for publication after revising in the following aspects:

--The combined the grain size factor and the Sc2O3 concentration on the stabilization of cubic phase should be concluded quantitatively.

--In section 3.3, the morphology of films deposited at RT as well as annealing was included. It is recommended to modify the subheading accordingly.

Author Response

We would like to thank the reviewer for the constructive and valuable comments. Many thanks for your time and consideration. We hope that the changes we have made will improve our article.  

Reviewer 2

Comments and Suggestions for Authors

Review of nanomaterials-2940208, entitled "Nanocrystalline cubic phase scandium-stabilized zirconia thin films".

The current manuscript focuses on the size factor and the dopant concentration range for the formation and stabilization of the cubic phase in scandium-stabilized zirconia (ScSZ) films. It is well-organized and written, it would be great helpful for understanding the stabilization of ScSZ. It is recommended to be accepted for publication after revising in the following aspects:

--The combined the grain size factor and the Sc2O3 concentration on the stabilization of cubic phase should be concluded quantitatively.

Thank you for your comment. The article deals with the influence of crystallite size (coherent scattering region, CSR) on the formation and stabilization of cubic zirconia. In general, CSR size is smaller than the grain size (depicted using electron microscopy), since the CSR region corresponds to the ordered region of grain and does not include distorted boundaries. As shown in section 3.3, the grain size of the co-sputtered ScSZ films depends mostly on their annealing temperature and conditions. Whereas an increase in the Sc2O3 concentration (at a constant substrate temperature) leads to a change in the crystallite size, and, as can be seen from Fig. 4, the dependence is not linear. The process of formation and stabilization of the cubic phase is affected not only by the size of crystallites and dopant concentration but also by oxygen vacancies, the number of which, in turn, depends on both the dopant concentration and the crystallite size. Note that experimental techniques and conditions are also essential players in c-phase stabilization. Correct separation of these factors is a complicated task because many parameters that mutually influence each other must be modified and analyzed, and solving this problem was not the purpose of the study.

--In section 3.3, the morphology of films deposited at RT as well as annealing was included. It is recommended to modify the subheading accordingly.

Thank you for your comment. The subhead is changed:

3.3 Morphology of the as-deposited and annealed RT co-sputtered ScSZ films

Submission Date

14 March 2024

Date of this review

26 Mar 2024 03:31:01

Reviewer 3 Report

Comments and Suggestions for Authors

Comments to Manuscript--Nanomaterials-2940208

This paper presented the relationship between the size factor and the dopant concentration range for the formation and stabilization of the cubic phase in scandium-stabilized zirconia (ScSZ) films. The thin films of (ZrO2)1-x(Sc2O3)x, with x from 0 to 0.2, were deposited on room-temperature substrates by reactive direct current magnetron co-sputtering. The crystal structure of films with average crystallite size of 85 Å was cubic at Sc2O3 content from 6.5 to 17.5 mol%, which is much broader than the range of 8–12 mol.% of the conventional deposition methods. There are some comments toward the research.

1.     To study the stabilization of the cubic phase in scandium-stabilized ScSZ films, the annealing temperature is 550 and 1100 °C, respectively. Why choice the two temperatures?

2.     It is suggested to add the methods of the obtaining lattice parameter and crystallite size from the XRD spectrum.  

3.     There is error bar in each data of the lattice parameter and crystallite size as shown in figure 4, 5 and 6. How many samples(or XRD spectrum) do you take for each concentration of Sc2O3, mol%?

Author Response

We would like to thank the reviewer for the constructive and valuable comments, as well as for your time and consideration. We hope that the changes we have made will improve our article.

Reviewer 3

Comments and Suggestions for Authors

Comments to Manuscript--Nanomaterials-2940208

This paper presented the relationship between the size factor and the dopant concentration range for the formation and stabilization of the cubic phase in scandium-stabilized zirconia (ScSZ) films. The thin films of (ZrO2)1-x(Sc2O3)x, with x from 0 to 0.2, were deposited on room-temperature substrates by reactive direct current magnetron co-sputtering. The crystal structure of films with average crystallite size of 85 Å was cubic at Sc2O3 content from 6.5 to 17.5 mol%, which is much broader than the range of 8–12 mol.% of the conventional deposition methods. There are some comments toward the research.

  1. To study the stabilization of the cubic phase in scandium-stabilized ScSZ films, the annealing temperature is 550 and 1100°C, respectively. Why choice the two temperatures?

Thank you for your comment. An explanation has been added to the Introduction, page 5, and is highlighted in yellow:

Therefore, the influence of subsequent annealing at 550 and 1100°C (low-temperature SOFCs operating temperature and maximum sintering temperature of SOFC functional layers, respectively) in the air on the structure, lattice parameters, and morphology of ScSZ sputtered thin films was also studied.

  1. It is suggested to add the methods of the obtaining lattice parameter and crystallite size from the XRD spectrum.

We would like to draw the attention of the respected reviewer that methods for determining the lattice parameter and crystallite size from the X-ray spectrum are presented in the sample characteristics section 2.2, page 6:

The structural characteristics of the co-sputtered ScSZ films were determined with an X-ray powder diffractometer (SmartLab SE, Rigaku, Japan) with Cu-Kα radiation (λ = 1.5460 Å). The XRD patterns were registered in 2θ geometry (angle of incidence 3°) in the range of 20–80° with a step of 0.03° and a rate of 0.5°/min. The phase analysis was performed using SmartLab Studio II ver. 4.2.44.0, using the Rietveld Refinement method in the 20–70° range. The crystallite size and lattice parameter were calculated using the Comprehensive Analysis module (of the same program) and the Halder-Wagner modeling method.

  1. There is error bar in each data of the lattice parameter and crystallite size as shown in figure 4, 5 and 6. How many samples (or XRD spectrum) do you take for each concentration of Sc2O3, mol%?

The SmartLab Studio II software calculated the error bars for the lattice parameter and crystalline sizes in the process of the Rietveld Refinement and Comprehensive Analysis of the diffraction patterns. These error bars are presented in Figs. 4, 5, 6.

Regarding the total number of samples examined, approximately 200 X-ray diffraction patterns were recorded and processed for more than 160 ScSZ samples of varying concentrations during the study.